# Seeing the forest and the tree:
# Building representations of both individual and collective dynamics with transformers

**Ran Liu,**[*] **Mehdi Azabou,  Max Dabagia,  Jingyun Xiao,  Eva L. Dyer**[*]
Georgia Institute of Technology

## Abstract

Complex time-varying systems are often studied by abstracting away from the dynamics of individual components to build a model of the population-level dynamics from the start. However, when building a population-level description, it can be easy to lose sight of each individual and how they contribute to the larger picture. In this paper, we present a novel transformer architecture for learning from time-varying data that builds descriptions of both the individual as well as the collective population dynamics. Rather than combining all of our data into our model at the onset, we develop a separable architecture that operates on individual time-series first before passing them forward; this induces a permutation-invariance property and can be used to transfer across systems of different size and order. After demonstrating that our model can be applied to successfully recover complex interactions and dynamics in many-body systems, we apply our approach to populations of neurons in the nervous system. On neural activity datasets, we show that our model not only yields robust decoding performance, but also provides impressive performance in transfer across recordings of different animals without any neuron-level correspondence. By enabling flexible pre-training that can be transferred to neural recordings of different size and order, our work provides a first step towards creating a foundation model for neural decoding.

## 1   Introduction

Complex systems (such as the brain) contain multiple individual elements (e.g. neurons) that interact dynamically to generate their outputs. The process by which these local interactions give rise to large-scale behaviors is important in many domains of science and engineering, from ecology [1, 2] and social networks [3–5] to microbial interactions [6] and brain dynamics [7].

A natural way to model the activity of a system is to build a collective or population-level view, where we consider individuals (or channels) jointly to determine the dynamics of the population (Figure 1(A)). In many cases, studying systems from this population-level perspective has provided important insights into collective computations and emergent behaviors [7]; however, it is also possible to lose sight of the contributions of different individuals' dynamics to the final prediction or inference. This is important in many settings where the dynamics of different individuals may be of interest, either due to their different functional roles in the system [8, 9, 6, 10], or due to shift in their dynamics because of sensor displacement or corruption [11–13]. Moving forward, we need methods that can build good population-level representations while also providing an interpretable view of the data at the individual level.

---

[*]Contact: rliu361@gatech.edu, evadyer@gatech.edu. Project page and code: https://nerdslab.github.io/EIT/

36th Conference on Neural Information Processing Systems (NeurIPS 2022).

Here, we present a new framework for modeling time-varying observations of a system which uses dynamic embeddings of individual channels to construct a population-level view (Figure 1(B-C)). Our model, which we dub *Embedded Interaction Transformer* or EIT, decomposes population dynamics by first learning rich features from individual time-series before incorporating information and learned interactions across different individuals in the population. One critical benefit of our model is *spatial/individual separability*: it builds a population-level representation from embeddings of individual channels, which naturally leads to channel-level permutation invariance. In domain generalization tasks, this means a trained model can be tested with permuted channels or entirely different numbers of channels.

To first understand how the model captures different types of interactions and dynamics, we experiment with synthetic many-body systems. Under different types of interactions, we show that our model can be applied to successfully recover the dynamics of known systems.

We then turn our attention to recordings from the nervous system [14], where we have access to readouts from populations of neurons in the primary motor cortex of two rhesus macaques that are performing the same underlying center-out reaching motor task. The stability of the neural representations and collective dynamics found in data from these animals [14–16] makes them an ideal testbed to examine the generalization of our approach. We show that, remarkably, generalization not only occurs across recording sessions (with different sets of neurons) from the same animal, but that we can also transfer across animals through only a simple linear readout. The performance of the linear decoder based on pre-trained weights in the across-animal transfer outperforms many models that are trained and tested on the same animal. This result provides an exciting path forward in neural decoding across large and diverse datasets from different animals and highlights the utility of our architecture.

The main contributions of this work are as follows:

- In Section 3, we introduce *Embedded Interaction Transformer* (EIT): a novel framework for learning from multi-variate time-series data that decomposes the dynamics of individual channels and their interactions through a two-stage transformer architecture.

- In Section 3.3, we propose methods for generalization across datasets of different input size (number of channels) and ordering. We show that by decomposing individuals and interactions, our architecture can be used to find functional correspondence between channels in different datasets by measuring the similarity in their embeddings with the Wasserstein divergence.

- In Section 4, we apply EIT to both many-body systems and neural activity recordings. After demonstrating our model's robust decoding performance, we validate its ability to transfer individual dynamics by performing domain generalization across different neural recordings and investigate the alignments of neurons across different populations.

## 2 Background and Related Work

### 2.1 Transformers

**Self-attention.** Transformers have revolutionized natural language processing (NLP) through the mechanism of self-attention [17], which helps the model to learn long-range interactions across different elements (represented by tokens) in a sequence. Consider a sequence $X = [\mathbf{x}^1 \ \mathbf{x}^2 \ \ldots \ \mathbf{x}^N]^T \in \mathbb{R}^{N \times d}$ consisting of $N$ tokens of $d$-dimensions. The self-attention mechanism is built on the notion of queries $Q$, keys $K$, and values $V$, which are linear projections of the token embeddings. Let $Q = XW_Q, K = XW_K, V = XW_V$, where $Q \in \mathbb{R}^{N \times d_q}$, $K \in \mathbb{R}^{N \times d_k}$, $V \in \mathbb{R}^{N \times d_v}$. The attention operation can be written as:

$$\text{Attention}(Q, K, V) = \text{softmax}\left(\frac{1}{\sqrt{d_k}}QK^T\right)V, \tag{1}$$

where $\text{softmax}(\cdot)$ denotes a row-wise softmax normalization function.

**Multi-head self-attention (MSA).** Rather than only learning one set of keys, queries, and values for our tokens, MSA allows each head to find different patterns in the data that are useful for inference. Each of the $h$ heads provides a $d_v$ dimensional output, and the outputs from all heads are concatenated into a $d_v \cdot h$ vector, which is further projected to produce the final representation.

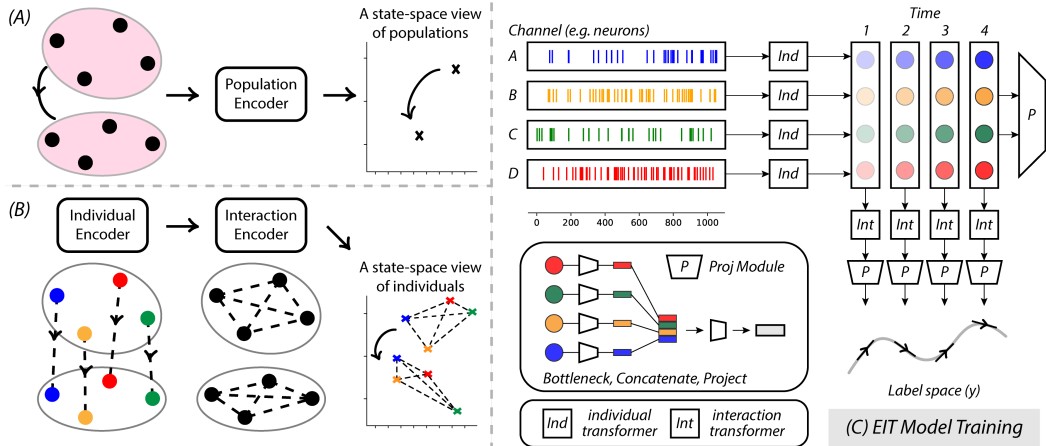

**Figure 1:** *Embedded Interaction Transformer (EIT).* **(A)** A traditional state-space view would treat the collective dynamics as a population right from the beginning, and use a population encoder to learn how the dynamics progress along time. **(B)** EIT learns dynamic embeddings of each channel with an individual encoder at the beginning. After embedding each channel's dynamics, we feed them into an interaction encoder to build a population representation. The two encoders work together to build a representation space that is richer than that of the traditional method. **(C)** The detailed architecture: EIT consists of an individual transformer that processes data for each individual, an interaction transformer that processes embeddings at each timepoint, and two projection modules at the end of both transformers.

The full operation of a transformer of $L$ layers can be written as below:

$$
\begin{aligned}
Z_0 &= \left[ \text{Embed}(\mathbf{x}^1), \text{Embed}(\mathbf{x}^2), \cdots \text{Embed}(\mathbf{x}^N) \right] + \mathbf{E}_{\text{pos}} \\
Z_{\ell+1} &= Z_\ell + \text{MSA}(Z_\ell) + \text{FF}(Z_\ell + \text{MSA}(Z_\ell)), \; \ell = \{0, \ldots, L-1\}
\end{aligned}
\tag{2}
$$

where $Z_0$ is the summation of the individual embedding of each data token and the positional embedding $\mathbf{E}_{\text{pos}}$ that helps to retain the positional information, and each transformer layer is the combination of the MSA operation ($\text{MSA}(\cdot)$), the projection ($\text{FF}(\cdot)$), and residual connections.

**Transformers for multi-channel time-series.** Multi-channel time-series are a natural candidate for modeling with the transformer architecture. One common approach of modeling many time-varying data streams with transformers is to first aggregate features across all channels into a combined representation at the beginning of the model. The population dynamics is thus learnt with the resulting embedding via a temporal transformer for the purpose of inference. There are many complex ways to create this embedding: [18] and [19] extract embeddings from multivariate physical systems with Koopman operators before feeding the resulting representation into a temporal transformer; [20] use a graph neural network to embed interconnected-structures to perform skeleton-based action recognition; [21] use a convolutional architecture to extract image embeddings before feeding them into a transformer.

Another approach is to re-design the attention block such that the attention operation is computed both along the temporal and spatial dimension. Many variants of this 'spatial-temporal' attention block have been shown effective: [22] propose a non-autoregressive module to generate queries for time series forecasting; [23] use stacked spatial-temporal modules for traffic flow forecasting; GroupFormer [24] embed spatial and temporal context in parallel to create a clustered attention block for group activity recognition. While our approach also makes use of the high-level idea of separating spatial (individual) and temporal information, crucially, we restrained the direct computation between the spatially-related attention map and the temporally-related attention map, which yields a representation of the individual which is completely free of spatial interactions.

**Spacetime attention in video transformers.** With the advances of the Vision Transformer [25] as a new way to extract image embeddings, many 'spatial-temporal transformer' architectures have been developed in the video domain [26–28]. Such works explore and propose interesting solutions for how to organize spatial attention and temporal attention with either coupled (series) [28] and factorized (parallel) attention blocks [26], as well as how to create better tokens for videos by creating three-dimensional spatio-temporal 'tubes' as the tubelet tokenizations [26]. However, these methods

leverage inductive biases that are specific to images and videos, and do not process potentially separatable channels that we are interested in.

**Object-centric representation learning.** There are many representation learning methods in video [29–32], physics-guided complex systems [33], and behavior modeling [34, 35] that also aim to learn interactions between discrete objects observed in the data. In these approaches, interactions between different objects are typically learned in a joint manner by combining the inferred representation of the objects into a common representation. In contrast, our framework aims to decompose dynamics into two parts, forming a representation of the dynamics of each individual source or object in addition to a representation of the interactions across many sources. Thus, one could imagine using our approach to build enhanced representations of object dynamics in other vision and behavioral neuroscience applications [30, 34]. It would be interesting to see if an enhanced or individual representation could be used to further improve the performance of upstream object localization and inference approaches.

## 2.2 Neural decoding and the challenge of generalization

The brain is an incredibly complex system. Neural activity is guided both by a tight interplay of membrane dynamics and ionic currents within a single neuron [36–39] as well as interactions with other neurons over larger distributed circuits [40–42]. Understanding how populations of neurons work together to represent their inputs is an important objective in modern neuroscience.

Accordingly, constructing representations that can explain neural population activity is an active area of research with many existing approaches. Discriminative models build representations that align or predict temporally-adjacent and masked neural activity, and include work such as MYOW [43] and the Neural Data Transformer (NDT) [44]. Generative approaches instead attempt to model neural activity as driven by latent factors [45], with success using switching linear dynamical systems [46] and recurrent neural networks [47]. Recently, SwapVAE [48] emerged as a hybrid of these two approaches, combining a latent factor model of neural activity with self-supervised learning techniques to predict across data augmentations. However, all of these methods produce representations exclusively at the population level.

**The challenges of transfer across recordings of neural population activity.** More often than not, neural recordings are collected from different sets of neurons within a brain region of interest. As a result, it is very difficult to generalize across neural recordings and the integration of many neural datasets remains a critical and yet open problem [49, 50].

Many existing approaches attempt to address this challenge by learning population-level descriptions of neural data and finding ways to align their latent representations [50, 14]. For example, [51] proposed a manifold alignment method to continuously update a neural decoder over time as neurons shift in and out of the recording; [52] proposed an adversarial domain adaptation for stable brain-machine interfaces; [53] proposed a robust alignment of cross-session recordings of neural population activity through unsupervised domain adaptation; [47] fit each set of neurons with its own encoder and used dynamics to inform alignment in the latent space during training. While such approaches have provided useful insights into the stability of neural representations across subsets of distinct neurons in the same and different brains solving the same task [16], the representations used for alignment are all formed jointly across the entire neural recording and thus each new dataset needs a different encoder or unique re-mapping, making it inherently unscalable when applied to many new datasets. EIT provides a novel strategy for decomposing neural dynamics that can be applied to arbitrary numbers of new recordings, making it possible to scale to large numbers of datasets. In addition, EIT can potentially be used to find neuron-level correspondence (or functional correspondence) through its decomposed representation of the population-level activity.

## 3 Methods

### 3.1 Formulation and Setup

In this work, we consider time-varying prediction from multi-variate time-series observations. Consider $X \in \mathbb{R}^{N \times T}$ as an input datum that measures the dynamics of $N$ individual channels over $T$ points in time. Let $x_{ij}$ denote the value of channel $i$ at the $j$-th time point, where $1 \leq i \leq N$ and

$1 \leq j \leq T$, and let $X^{(i)}$ denote the time series of the $i$-th channel. We consider time-varying labels $y = \{y_1, \ldots, y_T\}$ where $y_j \in \mathbb{R}^d$ is the label at the $j$-th time point.

Our aim is to find a function which takes input $X$ to predict target variables $y$. Instead of jointly considering individual time-series immediately, we instead decompose the model into two functions $f : \mathbb{R}^T \to \mathbb{R}^T$ and $g : \mathbb{R}^N \to \mathbb{R}^N$, so that the optimization problem becomes:

$$\min_{f,g,\mathrm{Proj}} \ \mathbb{E}_X \left[ \sum_{j=1}^{T} \ell \left( \ \mathrm{Proj} \left[ g \left( f(X^{(1)})_j, \ldots, f(X^{(N)})_j \right) \right], \ y_j \ \right) \right] \tag{3}$$

where $\mathrm{Proj}$ projects latent representations to predict the labels, and $\ell(\cdot, \cdot)$ is a loss measuring the discrepancy between the labels and the time-varying output from the decoder. In words, $f$ is applied to each time-series independently, while $g$ combines individual embeddings across the population at a single time step. Crucially, this decomposition provides $f$ invariance to the number of channels fed into the model. In practice, we map each $x_{ij}$ to a higher dimension to increase capacity for inference.

### 3.2 Approach

As shown in Figure 1(B), `EIT` decomposes collections of temporal data through two transformers. The first learns representations of the dynamics of individual channels and the second learns their population-level interactions.

**A temporal-spatial module that disentangles individual dynamics and their interactions.** To separate interactions at the population level from the individual dynamics, we start by building representations from individual time-series. For the $i$-th channel, we obtain an initial embedding of $T$ temporal tokens as $Z_0^{(i)} = [\mathrm{Embed}(x_{i1}), \mathrm{Embed}(x_{i2}), ..., \mathrm{Embed}(x_{iT})]$, where $\mathrm{Embed} : \mathbb{R} \to \mathbb{R}^M$ and $Z_0^{(j)} \in \mathbb{R}^{T \times M}$. We apply a transformer with multi-head attention to the resulting sequence using Equation (2). Let $\widehat{Z}^{(i)} \in \mathbb{R}^{T \times M}$ denote the final embeddings with elements $\hat{z}_{ij}$ obtained from the first transformer in our model and $\widehat{Z} \in \mathbb{R}^{N \times T \times M}$ be the combined embedded output.

After building representations of dynamics at the scale of individual time-series, we then pass the representations into a population-level transformer to capture interactions between all of the individual time-series. To do this, we take the output of the first module $\widehat{Z}$ and slice it along the temporal axis to yield an embedding of each time point in terms of the different channels. Denote this new sequential reslicing of the data at the $j$-th timepoint as $V_0^{(j)} = [\hat{z}_{1j}, \hat{z}_{2j}, ..., \hat{z}_{Nj}] + \mathbf{E}_{\mathrm{pos}}$, where $\mathbf{E}_{\mathrm{pos}} \in \mathbb{R}^{N \times M}$ is the fixed positional embedding that is dependent on neuron $i$'s identity. After applying the attention block operations in Equation (2) again, we then arrive at our final embedding of our time-series as $\widehat{V} \in \mathbb{R}^{N \times T \times M}$. Let $\hat{v}_{ij}$ be the final representation of the observation $x_{ij}$.

**A projection module that preserves individual identity.** To form a population representation from the embeddings of individual channels, we design a projection module that preserves individual identities. The projection module $\mathrm{Proj}(\cdot)$ consists of two parts: 1) For representations of individuals $z_{ij} \in \mathbb{R}^M$ for the $i$ individual and $j$-th time-point, $\mathrm{Proj}$ first bottleneck the individual latents with a function $bottleneck : \mathbb{R}^M \to \mathbb{R}$ to reduce the dimensions. 2) To form a population representation, $\mathrm{Proj}$ then concatenates the representation across individuals at the $j$-th time point, and learns another projection $project : \mathbb{R}^N \to \mathbb{R}^d$ to infer the label $y_j$. Through this operation, we could obtain both the population representation that could be used for inference and the individual representation that is affected in the minimal level.

**Learning representations through a multi-stage loss.** To build an unified representation that can provide both a description at the individual-level and the population-level, we compose a loss function as follows:

$$\mathcal{L}_{total} = \sum_{j=1}^{T} \ell(\mathrm{Proj\text{-}Int}([\hat{v}_{1j}, \ldots, \hat{v}_{Nj}]), y_j) + \alpha \cdot \ell(\mathrm{Proj\text{-}Ind}([\hat{z}_{1j}, \ldots, \hat{z}_{Nj}]), y_j), \tag{4}$$

where $\mathrm{Proj\text{-}Int}(\cdot)$ and $\mathrm{Proj\text{-}Ind}(\cdot)$ denote two projection modules after the population and individual transformers, respectively; $\ell$ can be set to either classification loss (CrossEntropy) or regression loss (MSE), depending on the type of prediction target we consider; $\alpha$ is a scaling factor that determines

how much emphasis is placed on prediction from individual representations in the first half of the network.

*Remark:* By setting $\alpha > 0$, we train a projection module on the intermediate individual-level embeddings that provides a purely individual-based representation of the population. Empirically, this more flexible architecture also seems to improve the stability of training when compared with setting $\alpha = 0$. Note that the first part of Equation 4 is a special case of Equation 3 when both the individual module and interaction module are transformer encoders.

### 3.3 Generalization across domains through linear decoding and functional alignment

A key element of our framework is that, after training, we can use the first part of the network ($f$) that we have learned for individual-level dynamics, on a new dataset of arbitrary size (number of neurons) and ordering. Here, we describe: (i) the linear probing approach for transfer across different sized or shuffled populations, and (ii) ways to characterize alignment of functional properties of different individual channels that are studied.

**Decoding from a new population of different dimension and ordering.**   In many domains, it is difficult to reliably preserve channels across datasets. Our model instead provides a flexible framework for domain generalization as the first transformer ($f$) operates only on individual time-series without considering the entire population. Specifically, to transfer to a new dataset, we can learn a new projection $h(\cdot)$ that acts on the concatenated pre-trained embeddings (outputs of $f$) to build predictions about downstream target variables $y$:

$$h^* = \arg\min_{h} \ \ell(h([\widehat{z}_{1j}, \ldots, \widehat{z}_{Nj}]), y_j), \tag{5}$$

where $h$ is a decoder mapping representations to predicted labels, $\widehat{z}_{ij}$ are the representations formed for the $i$ individual at the $j$-th time point, and $y_j$ is the label at $j$-th time point. When we restrict $h$ to be linear, this approach is similar to the evaluation methods in self-supervised learning [54, 55], where a linear decoder is used to evaluate the quality of the frozen representation based on various different downstream tasks.

**Functional alignment through Wasserstein-based representational similarity analysis.**   Our model also provides a flexible framework for identifying correspondences that might exist between channels in different recordings. For instance, in the case of neural recordings, we can use EIT to find correspondence across neurons in different recording sessions. Recall that for each individual time-series $X^{(i)} \in \mathbb{R}^T$, our model builds a representation space of size $\widehat{Z}^{(i)} \in \mathbb{R}^{T \times M}$. To characterize the similarity across different channels, we pass many samples through the individual encoder to estimate the distribution of $\widehat{Z}^{(i)} \in \mathbb{R}^{T \times M}$. We denote this distribution of individual $i$ as $R_i$.

We then use the Wasserstein divergence ($\mathcal{W}$), a measure of distributional distance motivated by optimal transport (OT) [56, 57] to obtain robust measures of similarity (More details in Appendix 1.1). For one set of distributions $\{R_1^{(1)}, R_2^{(1)}, \ldots, R_{N_1}^{(1)}\}$ from domain (1) and another set of distributions $\{R_1^{(2)}, R_2^{(2)}, \ldots, R_{N_2}^{(2)}\}$ from domain (2), we compute the divergence between all pairs of individuals, which yields a matrix $D \in \mathbb{R}^{N_1 \times N_2}$ where $D[i][j] = \mathcal{W}(R_i^{(1)}, R_j^{(2)})$.

## 4 Experiments

### 4.1 Synthetic experiment: observing superposed many-body systems

We first tested our model on synthetic many-body systems [58] (see [59, 60] for other possible synthetic datasets with additional complexity). As shown in Figure 2(A), for a many-body system, a body's observed trajectory is decided by its own properties (its mass and starting point), and its own intended dynamics are further affected by its interaction with other bodies. Furthermore, for certain many-body systems (e.g. $k$-body systems for $k \geq 3$) the trajectories are chaotic in nature, which is aligned with the stochasticity of neural activity [61, 62]. To test whether our model is able to capture, disentangle, and decode the body dynamics, we create an experiment of 'superposed' many-body systems (as shown in Figure 2(B)), where two separate many-body systems are overlaid with each other as to create a system where some connections exists (bodies within the same system), while some do not (bodies from different systems).

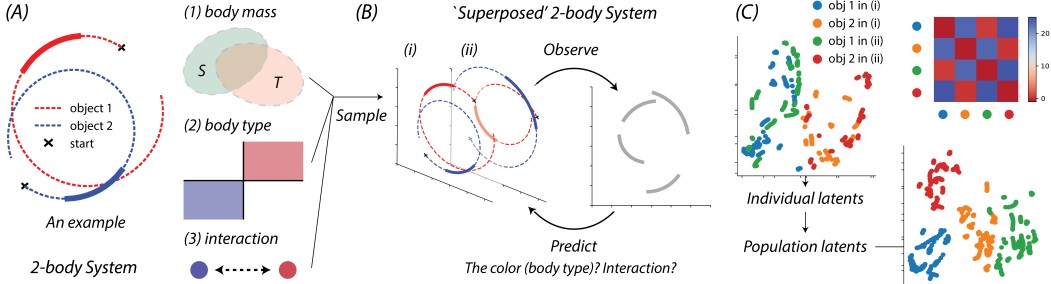

**Figure 2:** Synthetic Experiments. (A) We show an example of a two-body system, where the system is roughly controlled by three factors: the body mass, the body type, and the body interaction. (B) We create a 'superposed' two-body system by sampling initial conditions from the distribution, observing the produced trajectories over shorter time intervals, and predicting the body type and their interaction from the trajectories. (C) We visualize both the individual latent space and the population latent space. The upper right corner shows how our OT evaluation metric can align and distinguish bodies of the same type (red denotes lower discrepancy).

**Generating non-chaotic and chaotic systems.**   We generated non-chaotic two-body systems and chaotic three-body systems in a relatively stable near-circular way [63], where the trajectories are obtained with the Explicit Runge-Kutta method [64]. We refer the readers to Appendix 2.1 for the details of the set of second-order differential equations that guide the movement.

As shown in Figure 2(B), we create the 'superposition' of many-body systems in the following pipeline: two independent many-body systems are *sampled*, and then their trajectories without additional type or interaction information are *observed*. When training the model, we aim to generate a latent representation for each body at each time that *predicts* the underlying factors that guide the body movements. Note that we use a different distribution of body mass to separate the training set (Source) and testing set (Target), which makes the generated body trajectories purely controlled by two factors: 1) where the body initially starts (as the body **types**); and 2) which body is this body interacting with (as the **interaction**). The model is trained to recover these two factors.

For the two-body system and the three-body system, we sample 10 consecutive observations with a gap of $0.2$ and 50 consecutive observations with a gap of $0.05$ within time span $[0, 10]$, respectively. For each system that consists of different body types and interactions, we generate 500 trials to provide large variability of the movements. We performed a $80/20\%$ train/test split throughout, where the testing set contains unseen combinations of body weights that determine the trajectory.

**Model performance.**      We evaluate the performance of `EIT` by investigating its ability to decode the body types ('Type'), body interaction ('Int'), as well as all possible combinations ('All') of them. The experiments are performed under the domain generalization scheme where the body mass distribution of the testing set (T) is different from that of the training set (S).

**Table 1:** *Decoding performance of `EIT` on many-body systems when compared to a baseline model (BM).*

|  | 2-body | | 3-body | |
|---|---|---|---|---|
|  | BM | EIT | BM | EIT |
| 'Int' | 93.97 | 97.28 | 19.99 | 91.70 |
| 'Type' | 97.40 | 99.93 | 4.03 | 37.76 |
| 'All' | 87.90 | 94.72 | 2.03 | 41.99 |

We benchmark `EIT` with a transformer [44] with the same depth and amount of heads that considers the population (create the population representation) right from the start of the model. As shown in Table 1, `EIT` consistently outperforms the baseline model (BM) in all cases, where our model showed significant advantage over the baseline when predicting $\approx 100$ classes (the 'Type' and 'All' in three-body setting are 90 and 360 classes, respectively). The advantages of the architecture are especially apparent when it comes to the three-body chaotic systems: as the system trajectories provide much more variability and are highly sensitive to the initial conditions (as body types), our model provides significantly better performance by analyzing the individual dynamics separately to provide a stable representation space for each individual.

We visualized both the individual and population latent space in Figure 2(C) for the two-body systems. Our individual-module successfully distinguishes different types of bodies, while the population-module further separates different systems by learning the interaction of bodies. We refer readers to Appendix 2.1 for visualizations of the three-body systems. On the top-right corner, we show the OT

**Table 2:** *Performance on behavioral decoding from populations of neurons in the motor cortex.*

*I. Decoding performance on neural datasets*

| | Mihi-Chewie ($T=2$) | | | | Mihi-Chewie ($T=6$) | | | | Maze ($R^2$x100) | |
|---|---|---|---|---|---|---|---|---|---|---|
| | C-1 | C-2 | M-1 | M-2 | C-1 | C-2 | M-1 | M-2 | Vel | Pos |
| MLP | 74.22 | 74.54 | 78.17 | 74.42 | 78.91 | 90.74 | 87.90 | 84.11 | 60.64 | 78.59 |
| GRU | 75.78 | 75.46 | 79.96 | 74.22 | 84.72 | 90.12 | 85.98 | 78.81 | 79.97 | 94.01 |
| NDT | 59.90 | 58.56 | 73.02 | 70.74 | 64.93 | 63.73 | 80.56 | 71.96 | 76.01 | 90.07 |
| NDT-Sup | **80.47** | 80.56 | 83.93 | 80.79 | 87.33 | **94.29** | **96.83** | 91.47 | 82.02 | **94.88** |
| EIT (T) | 76.04 | 81.25 | 81.15 | 71.71 | 83.33 | 88.27 | 93.65 | 86.82 | 71.18 | 87.09 |
| EIT | 79.69 | **82.41** | **86.51** | **81.61** | **88.36** | 92.59 | 95.24 | **91.57** | **82.15** | 94.77 |

*II. Generalization performance - trained on one population, tested on another (more in Appendix.3.2)*

| | C-1 | C-2 | M-1 | M-2 | C-1 | C-2 | M-1 | M-2 | Vel | Pos |
|---|---|---|---|---|---|---|---|---|---|---|
| NDT$_{retrain}$(C-2) | 75.52 | - | 77.78 | 73.06 | 85.24 | - | 92.46 | 86.18 | × | × |
| NDT$_{retrain}$(M-2) | 74.48 | 70.37 | 76.78 | - | **86.28** | 89.20 | 91.99 | - | × | × |
| EIT (C-2) | **79.17** | - | 82.94 | **75.24** | 81.42 | - | 92.33 | **91.34** | × | × |
| EIT (M-2) | 78.13 | **81.02** | **84.13** | - | 84.72 | **91.06** | **93.25** | - | × | × |

distance matrix that is produced by our individual-based evaluation method on the test set. The matrix can clearly quantify the similarity between bodies, and reveals the different types of individuals.

## 4.2 Decoding behavior from neural populations

**Datasets.** The Mihi-Chewie reaching dataset [14] consists of stable behavior-based neural responses across different neuron populations and animals, and thus is used in previous methods for across-animal decoding [53, 52, 14] and neural representation learning [43, 48]. Mihi-Chewie is a spike sorted dataset where two rhesus macaques, Chewie ('C') and Mihi ('M'), are trained to perform a simplified reaching task to one of eight targets, while their neural activities in the primary motor cortex were simultaneously recorded. The dataset contains two recordings of different sets of neurons for each of the subjects, for a total of four sub-datasets. The Jenkins' Maze dataset [65] is a dataset in the Neural Latents Benchmark [45] that contains activity from the primary motor and dorsal premotor cortex of a rhesus macaque named Jenkins ('J'). In this dataset, J reaches towards targets while avoiding the boundaries of maze that appears on the screen. Since this dataset provides more complex behaviour trajectories with a complete collection of continuous labels (movement velocity and hand position), we used this dataset to examine if our model creates an individual representation space that is rich enough to decode continuous targets. All neural datasets are sorted and binned into 100 ms intervals by counting the number of spikes each neuron emits during that time frame to generate a time-series for each neuron.

**Experimental setup.** Both the temporal and spatial transformer have a depth of 2 and 6 heads. We set the dimension of single neuron representation to be 16 through the transformer training, and bottleneck it to be 1d when evaluating the activity representation, which makes the activity representation equal to the total number of neurons. We train EIT end-to-end with an Adam optimizer of learning rate 0.0001 for 400 epochs. We benchmark our models' performance with a MLP, bi-directional GRU [66], NDT [44], and a supervised variant of the NDT (NDT-Sup) that we implemented for sequential data decoding. For domain generalization tasks, we re-trained the first-layer of NDT, and compared our model against numbers obtained when training on the same individuals. We also tested EIT 's performance when considering only non-sequential data and compared it with recent self-supervised methods [43, 48] in Appendix 3.2.

**Decoding performance.** As shown in Table 2, our model provides strong decoding performance on various benchmarks, both in terms of its classification of reach (Mihi-Chewie) and continuous decoding of position and velocity in the regression task studied (Jenkins). For the classification task on Mihi-Chewie, we followed [43, 48] and tested the model's robustness under both shorter sequence setting ($T$=2) and longer sequence setting ($T$=6), while for J's Maze we evaluated our model on the full sequence with a regression loss. Both our model and NDT [44] provide good performance, and in some cases, the EIT (T) temporal transformer does quite well on these tasks. Our results suggests that our model is competitive on these diverse decoding tasks from different neural populations in multiple animals.

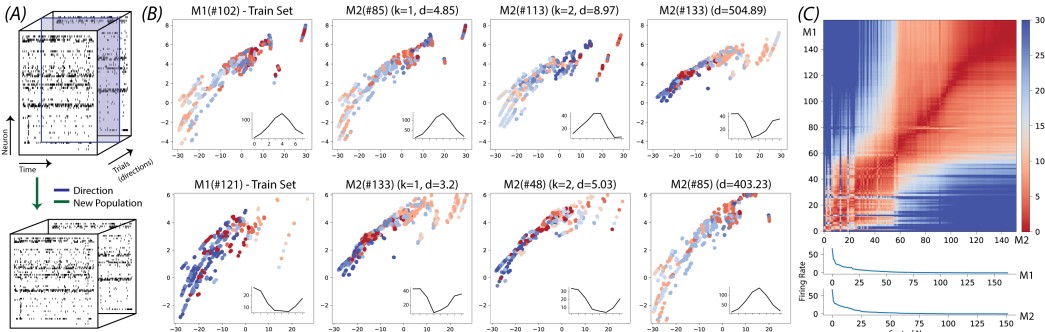

**Figure 3:** *Experiments on multi-neuron recordings from motor cortex.* (A) A schematic of the two transfer conditions considered for neural recordings. (B) Along each row (from left to right), we display a query neuron, its first and second nearest neighbor, and a high firing neuron that has large divergence (the divergence $d$ is also reported for all examples). The average firing rate per reach direction (tuning curve) is shown in the bottom right of each embedding. (C) The neuron-to-neuron Wasserstein divergences for M1-M2 encodes the similarity between the neuron-level embeddings (red denotes similar and blue is dissimilar). Below, we show the firing rates for the population along the rows (top) and columns (below) in descending order.

**Generalization task #1: across recording session and animal.** After validating the model's decoding performance, we investigate its generalization across different sessions and animals. As the number of neurons in different recordings varies, we cannot use the model in an end-to-end manner when testing on a new neural population of a different size. Instead, we train our model on one recording, freeze the weights, and apply the channel-invariant portion of our model to the new data as described in Section 3.3 (Figure 3(A)). In our experiments, we re-train only a final linear classification layer to predict the labels on the new dataset.

Even when tested on new populations of neurons, our model provides impressive overall performance in decoding across sessions and animals (Table 2, bottom). Here, we compare with a re-trained NDT (with a new input layer) and with models trained and tested on the same animal (within domain, top). In these cases, we find that through a linear readout from EIT , we can actually outperform many models that are trained from scratch within domain. These results suggests that our temporal transformer has learned information about the firing patterns of neurons that can be transferred across populations in the same and different animals.

To further investigate the underlying factors that contribute to our ability to generalize across populations, we visualized the latent space of individual neurons that have either a closer distribution (in terms of their OT distance) or distant distribution in Figure 3(B). In multiple cases, we found that the embedding of neurons in the training set (M1) had close neighbors in our test set (M2, same animal at a different time point) with similar tuning profiles (bottom right of each latent embedding). At the same time, neurons that were more distant had orthogonal (or distinct) functional tuning.

To characterize this functional alignment at a population level, we measured the divergence between all pairs of neurons in the train and test set (Figure 3(C)). When sorting neurons in both conditions by their firing rate (bottom), we found that the learned neuron representations have a good overall global correspondence in terms of their latent embeddings, which suggests that the learned latent representation of individuals contains sufficient information about neurons' overall firing rate. These experiments open up a lot of possibilities for finding functional groups in the individual embeddings and suggest that EIT could be used to find correspondence between neurons in different datasets.

**Generalization task #2: across behaviours.** To test the model's ability to transfer when the overall class distribution shifts between the train and the test, we trained on a limited amount of data with a limited set of targets (2 classes) and tested on all 8 classes. Again, we can test generalization in this condition by training a linear layer to decode from the individual transformer (trained on 2 classes) on the new test condition (full 8 classes). Our results

**Table 3:** *Performance on Mihi-Chewie reach decoding task when trained on two targets and tested on all eight targets.*

|         | C-1   | C-2   | M-1   | M-2   |
|---------|-------|-------|-------|-------|
| MLP     | 74.28 | 74.00 | 83.33 | 75.81 |
| GRU     | 74.64 | 70.67 | 78.49 | 79.30 |
| NDT-Sup | 75.72 | 71.00 | 84.41 | 82.80 |
| EIT     | **82.25** | **83.00** | **91.94** | **85.22** |

in Table 3 suggest that `EIT` can work well when tested in new behavioral conditions and provides significant gaps over other approaches, even when the training data is limited.

## 5 Conclusion

In this work, we introduced `EIT` as a model for learning representations of both individual and collective dynamics from multi-variate time-series data. In our experiments on both synthetic many-body systems and real-world neural systems, we demonstrated that our model not only provides state-of-the-art decoding performance, but also leads to impressive domain generalization thanks to its permutation-invariant design.

While our model provides a novel strategy for decomposing complex dynamics, we note that there are also limitations and areas for future work:

- *The downsides of building individual embeddings:* Training an individual module at the beginning not only requires prior knowledge about the separation of the system [30], but also might restrict the model's representational capacity (also discussed in [67]) due to the limited size of the final representation space. One possible solution to both challenges is to stack multiple individual-interaction modules [23], which would sacrifice the spatial/individual separability of the individual module. Instead, one could combine the proposed framework with individual/object disentanglement methods to both decompose mixtures of sources and to learn richer representations for individuals.

- *Replacing transformers with task-specific architectures:* `EIT` utilized transformer encoders for both the individual and interaction modules. However, it is possible to replace the transformer encoders with a task-specific architecture to process specific types of interactions, such as graph-based encoders [68, 23] or recurrent encoders [69], which might improve performance on certain tasks. Additionally, the individual module of `EIT` models individual dynamics in a deterministic way, but in certain cases (e.g. multi-armed bandit tasks for neural activities), it might be more appropriate to model dynamics and interactions in a probabilistic manner (such as in [70]).

- *Reducing the need for labels through self-supervised training:* While the individual representations of `EIT` generalize reasonably well across animals in the neural activity experiments, the model currently relies on labels to guide this functional alignment. Moving forward, `EIT` could be further extended through training the whole network in a self-supervised way, perhaps using masking and completion tasks [71, 72] or contrastive approaches for neural activity [43, 48]. Self-supervised training not only would eliminate the dependency on labels, but also might improve generalization.

**How `EIT` provides a new paradigm that can help advance foundation models for brain decoding.** In many domains, the ability to integrate large amounts of data from different sources into a single pre-trained model has enabled impressive advances on many downstream tasks [73–75]. With such a model for neural data analysis, we could similarly build powerful decoders that could leverage the large amounts of open neural data currently being generated [76, 77] to decode behaviors in diverse contexts and complex tasks. Here we show that by decoupling our learning into two stages and building an encoder that processes single neurons first, we can apply the front-end of our model to new collections of neurons without any modifications or re-training regardless of the mismatch of the inputs. We see this as a significant first step towards building a foundation model for neural decoding that can learn from diverse sets of neurons in different brains and contexts.

## Acknowledgements

This project was supported by NIH award 1R01EB029852-01, NSF award IIS-2039741, the NSF Graduate Research Fellowship Program (GRFP) for MD, as well as generous gifts from the Alfred Sloan Foundation, the McKnight Foundation, and the CIFAR Azrieli Global Scholars Program.

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
