# Appendix

## 1 Methods details

### 1.1 Functional mapping across domains with Optimal Transport (OT)

The Wasserstein divergence is defined as the amount of work required to warp one distribution $U \in \mathbb{R}^{N \times t}$ into another distribution $V \in \mathbb{R}^{N \times t}$. This mapping is done through what's referred to as a *transport plan*. Thus to compute the similarity between two distributions under this measure, we need to first solve the following optimization problem:

$$\mathcal{W}_\gamma(U, V) = \min \sum_{i,j=1}^{t} P_{i,j} C_{i,j} - \gamma H(P) \quad \text{subject to} \quad P \in \mathbb{R}_+^{t \times t}, P^T 1_t = 1_t, P 1_t = 1_t, \quad (6)$$

where $P_{i,j}$ is the transport plan and $C_{i,j}$ is the ground metric that measures the distance between point $i$ in the source and $j$ in the target. A common expression for the ground metric is to use either the L2 distance for the Mahalanobis metric which can be defined in terms of a matrix $L$ as $C_{i,j} = \|L(u_i) - L(v_j)\|_2^2$. The final term introduces some regularization into our objective by adding a entropy term on the transport plan. This will induce some smoothness and wiggle room in the solution of our objective.

### 1.2 Implementation of temporal transformer w/ offset attention.

In some settings (e.g. our experiments on neural datasets), the absolute value of the input may provide less information than the relative change of the input. Thus, following [78], we utilized the following offset self-attention inside the temporal transformer for learning:

$$Z_{l+1}^{(i)} = \text{MSA}(Z_l^{(i)}) + \text{FF}(Z_l^{(i)} + \text{MSA}(Z_l^{(i)})). \quad (7)$$

Note that in offset attention, we only use the MSA output and FF output but do not include $Z_\ell^{(i)}$ in our attention block computation.

## 2 Synthetic Experiments

### 2.1 More information about synthetic datasets

We modified the many-body systems initialization rules in [63] to create our own systems.

**Two-body system initialization** The trajectories are initialized in a near-circular way. The mass of the two bodies are sampled from range $[m_{min}, m_{max}]$. Both coordinates of the initial position of Type 1 body $(x_0^{(1)}, y_0^{(1)})$ is inside range $[0.5, 1.5]$, while the initial position of Type 2 body is $(x_0^{(2)}, y_0^{(2)}) = (-x_0^{(1)}, -y_0^{(1)})$. The initial velocity of each body is dependent on their initial positions, which is $(y_0/2r^{1.5}, x_0/2r^{1.5})$, with $r$ as the initial distance between the two bodies. To increase the diversity of the observed trajectories, we inject Gaussian noise ($\sigma = 0.05$) into trajectories by perturbing the initial velocities.

Since two-body systems are non-chaotic systems, we divide training set and testing set such that for training set $[m_{min}, m_{max}] = [0.8, 1.2]$, while testing set $[m_{min}, m_{max}] = [0.9, 1.3]$ to create domain distribution shifting. Aside from this, we made sure that the body mass combinations are different between the training set and testing set by re-using existing mass combinations in each set to create different systems.

**Three-body system initialization** For the chaotic three-body systems, we also apply initial condition regularization such that the initial trajectories of the system is also near-circular. Both coordinates of the initial position of Type 1 body $(x_0^{(1)}, y_0^{(1)})$ are sampled from $[0.9, 1.2]$, and the initial positions of Type 2 and Type 3 bodies are produced by rotating Type 1 body position by $120°$ and $240°$ degrees, respectively. The initial velocities of all bodies are based on their initial positions by rotating it by $90°$ and scaling it by $r^{1.5}$. We also inject Gaussian noise ($\sigma = 0.1$) into the initial velocities.

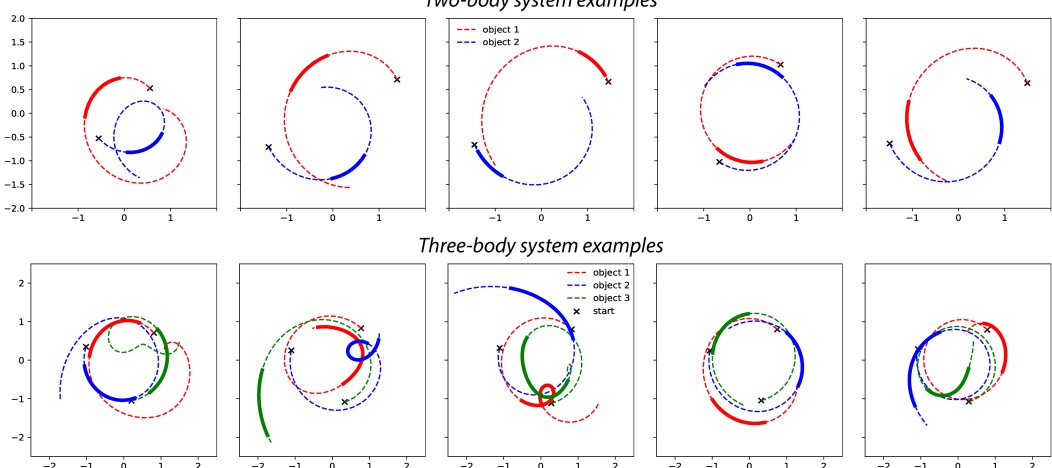

**Figure A1:** *Examples.* Examples of the two-body systems and three-body systems we generated.

Since there exists adequate chaotic behaviours in the resulting three-body systems (see figures below), we set $m_1 = m_2 = m_3 = 1$ across training and testing sets. We still re-use existing mass combinations in each set to create different systems to encourage domain shifting.

**Mathematical description of system motion**    Both systems are guided by the Newtonian equations:

$$\frac{d^2\mathbf{r}_i}{dt^2} = -\sum_{j \neq i} gm_j \frac{\mathbf{r}_i - \mathbf{r}_j}{|\mathbf{r}_i - \mathbf{r}_j|^3}$$

where $\mathbf{r}_i$ is the position of the $i$-th body, $m_j$ is the mass of the jth body, and $g$ is the gravitational constant (we set $g = 1$ across all experiments for simplicity).

While a general closed-form solution exists for two-body systems, three-body systems often give chaotic dynamics and thus require numerical methods to produce the trajectory. Given the initialization parameters defined above, we used the Explicit Runge-Kutta method [64] of order 5 to obtain system trajectory observations.

## 2.2    Many-body systems and latent space visualizations

We show a few examples of the two-body systems and three-body systems in Figure.A1. As we can see, the two-body systems are relatively stable and consistent, while three-body systems produce chaotic behaviours from time to time, which significantly increases the inference difficulty.

Similar as in Section.4.1, we show the latent space obtained from EIT trained on three-body datasets in Figure.A2. We can see that in the latent space of individuals, the representation discrepancy of bodies of the same type is relatively small. While in the latent space of population, the representation of different individuals is further separated and clustered into different groups. However, the chaotic nature of the three-body systems makes it difficult to obtain a perfectly clean latent space.

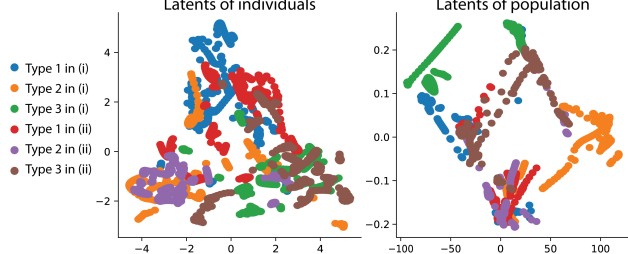

**Figure A2:** *Latent space visualization.* Latent space of individuals and population of the three-body systems.

# 3 Neural Experiments

## 3.1 More details on datasets and benchmark model implementations

### 3.1.1 Details on datasets

**Mihi-Chewie dataset**  The Mihi-Chewie dataset contains four separate recording from the primary motor cortex (M1). The recordings are collected from surgically implanted electrode arrays and are thresholded and spike sorted when collected. After binning the activities, we further compute the activity variance for each individual neuron and remove the ones with zero variance. After the pre-processing stages, there are 163, 148, 163, 152 neurons left for C-1, C-2, M-1, M-2, respectively.

**Jenkins' Maze dataset**  Jenkins' Maze contains recording from the primary motor and dorsal premotor cortices when a macaque is performing a delayed center-out reaches in a configured maze [65]. To perform experiments, we utilized the pre-processed data in [45], which contains the recording of 182 neurons. In addition to the neural activities, Jenkins' Maze dataset also contains behavioural labels such as the hand position and velocity of Jenkins, which are simultaneously recorded when Jenkins is performing the behavioural task. We averaged the behavioural label based on the same binning size as the regression labels for model training.

### 3.1.2 Details on model implementation

We include details on the model implementation as follows. Note that all latent dimensions are set to be consistent as the total amount of neurons. For all experiments, we use the Adam optimizer to optimize the model. All models are trained for 400 epochs for Mihi-Chewie experiments, and 600 epochs for J's Maze experiments.

- MLP: For both datasets, we used a multilayer perceptron consists of three cascaded blocks of fully-connected layer and ReLU activation layer. The model is trained with a learning rate of $0.001$.
- GRU: For both datasets, a single layer bi-directional gated recurrent unit RNN is used. The model is trained with a learning rate of $0.001$.
- NDT-Sup (Supervised NDT): For Mihi-Chewie experiments, we trained a 4-layer transformer, with 6 heads of dimension 32. The model is optimized with a learning rate of $0.0001$. For J's Maze experiments, we trained a 6-layer transformer with 8 heads of dimension 32, and optimized the model with a learning rate of $0.001$. We made sure that the architecture of NDT is in consistent with the architecture of EIT in terms of transformer complexity.
- NDT (Self-supervised NDT): While the network architecture of NDT remains the same as NDT-Sup, we performed masked autoencoding for self-supervised training: For each sample, we randomly mask $50\%$ time-steps of inputs and reconstruct the masked values with a reconstruction loss under the Poisson distribution assumption [47, 48]. During the inference stage, we freeze the model weights and train a linear layer to predict the downstream tasks. The resulting performance is used to indicate model's decoding quality.
- EIT : For the individual module (EIT (T)), we set the representation space of each single neuron as 16-dim, and used a 2-layer transformer for both experiments. The individual transformers have 6 heads of dimension 32 and 8 heads of dimension 32 for Mihi-Chewie dataset and J's Maze dataset, respectively. As for the interaction module: for Mihi-Chewie experiments, we used a 2-layer transformer with 6 heads of dimension 32 as consistent with the individual module. Note that in J's Maze experiments, we increased single neuron dimension to 256 in the interaction module with a 4-layer transformer of 8 heads with dimension 32. We found that increasing single neuron dimension plays a critical factor when performing the regression downstream tasks.

### 3.1.3 Details on hyperparameter tuning

For the decoding experiments: We did a grid-search of hyperparameters on the mihi-1 dataset, and used the same set of hyperparameters throughout all mihi-chewie experiments. Similar procedures are conducted for synthetic and Maze experiments, but the grid-search is done for each task.

For the generalization experiments: The training set (e.g. recordings of one animal) is divided into the training and validation sets based on the division in decoding experiments. The best-performing model parameters on the validation set are used for the testing set (e.g. recordings of another animal).

## 3.2 More experiments

**Additional generalization experiments**  In Section.4.2, we showed the performance of generalization across neurons and animals when the model is trained on C-2 and M-2. Here, we include the generalization results when the model is trained on C-1 and M-1. Interestingly and perhaps intuitively, we found that the generalization performance of `EIT` is dependent on the performance of the pre-trained model, as the model trained on C-1 gives worse performance in general.

**Table A1:** *Additional generalization performance.*

|  | *Mihi-Chewie (T = 2)* | | | | *Mihi-Chewie (T = 6)* | | | |
|---|---|---|---|---|---|---|---|---|
|  | C-1 | C-2 | M-1 | M-2 | C-1 | C-2 | M-1 | M-2 |
| EIT (C-1) | - | 81.25 | 83.53 | 74.42 | - | 87.04 | 91.93 | 86.18 |
| EIT (M-1) | 79.43 | 79.17 | - | 75.78 | 82.47 | 90.74 | - | 88.11 |

**Inference on single-time scale.**  In addition to testing the model's performance under the sequential setting, we are also interested in how the model performs when there exists no sequential data during inference, as in [43, 48]. Thus, we designed a single-time training/inference setting: Consider a data sequence $[x_1, \cdots, x_j, \cdots, x_t]$. During training, we either feed the model a copy of two data at a single-time point (e.g. $[x_j, x_j]$), or we mimic the temporal augmentation procedure, and fed the model a sequential data (e.g. $[x_j, x_{j+1}]$). During inference, we assume no temporal augmentation, and only feed the model copies of data. We used the output of the first datapoint as the output for inference.

**Table A2:** *Performance on Mihi-Chewie reach decoding task in single-time inference setting (T=1).*

|  | C-1 | C-2 | M-1 | M-2 |
|---|---|---|---|---|
| MLP | 63.54 | 65.28 | 67.66 | 65.70 |
| BYOL [55] | 66.65 | 64.56 | 72.64 | 67.44 |
| MYOW [43] | 70.54 | 72.33 | 73.40 | 71.80 |
| betaVAE [79] | 64.34 | 60.24 | 58.11 | 60.23 |
| SwapVAE [48] | 72.81 | 68.97 | 64.26 | 66.12 |
| GRU | 72.92 | 71.99 | 74.01 | 71.32 |
| NDT (Super) | 69.01 | 71.30 | 71.83 | 67.64 |
| **Ours (T)** | 70.31 | 74.54 | 74.60 | 67.83 |
| **Ours (T + S)** | **75.26** | **76.85** | **76.59** | **72.87** |

With restricted temporal context, we tested how well our model and other sequential models perform under this setting. As in Table A2, we compare with other methods that have been previously reported for single-time scale inference. Again, we show that our model, when essentially collapsed into a population-level model (with temporal 'augmentations'), outperforms other supervised and self-supervised models. However, we note that this training setting might introduce temporal information bias, as other sequential models (e.g. GRU) also provide better performance than previous benchmarks that are purely designed for single-time scale training and inference. Although this setting has its limitations, we present it as a promising future research direction for inference in the online setting.

**Generalization across time**  We also performed the generalization across time experiments. We divide the dataset such that the first $50\%$ data in time is used for training, while the rest $50\%$ data in time is used for testing. As shown in Table.A3, the temporal generalization performance is subpar in general, indicating that dynamics plays an important factor for the model's training. It remains unexplored whether the model would perform differently on free-behaviour datasets.

**Table A3:** *Accuracy (in %) for generalization across time.*

|  | C-1 | C-2 | M-1 | M-2 |
|---|---|---|---|---|
| Linear | 40.99 | 37.50 | 40.43 | 40.16 |
| GRU | 49.06 | 39.91 | 44.58 | 48.60 |
| NDT (Supervised) | 47.17 | 39.54 | 43.94 | 47.21 |
| EIT (T) | 39.94 | 33.06 | 42.98 | 45.50 |
| EIT | **51.57** | **42.31** | **45.14** | **50.16** |

**Method stability and sensitivity to the ordering of channels**  To examine our method's decoding stability across different random seeds, we repeated the Mihi-Chewie (T=2) decoding experiments over five random seeds, and computed the resulting mean accuracy and standard deviation as below

(the "without perm" row). Furthermore, to better understand the sensitivity of the method to channel ordering, we repeated our experiments with the whole model re-trained based on a permuted ordering of neurons for the same five different random seeds. The resulting mean accuracy and standard deviation are computed in the "with perm" row. The results suggest that EIT provides relatively stable performance across different random seeds and different orders of input channels.

**Table A4:** *Decoding stability and sensitivity on Mihi-Chewie (T=2) experiments.*

|  | C-1 | C-2 | M-1 | M-2 |
|---|---|---|---|---|
| Without perm | 79.86±1.23 | 82.13±0.52 | 86.83±0.76 | 80.41±0.92 |
| With perm | 80.22±1.79 | 81.84±1.88 | 85.78±0.98 | 80.05±1.70 |

### 3.3 Analysis of functional motifs in neuron-level representations

To find matchings between neurons and create the components that went into Figure 3, we did the following.

- (1) *Fully unsupervised OT:* Compute the pairwise OT matrix for all neurons in a source and target dataset. The similarity matrix is $N_1 \times N_2$.

- (2) *Partially supervised OT:* For each neuron, select all the time points from the trials over which a neuron has its strongest response (preferred direction) and use these points to form the neuron-level distribution. Then compute the pairwise OT matrix for all neurons in a source and target dataset. The similarity matrix is $N_1 \times N_2$.

- (3) *Nearest Neighbor Matching:* To find putative alignments across neurons in two datasets, we can use the nearest neighbor between a set of source and target neurons. To generate the matches in Figure 3(B), we use the label-assisted OT matrix to find correspondences. Here, we find that both the nearest and second nearest neighbors have similar overall tuning curves (shown on bottom right of embeddings). We also select the pair of neurons that the farthest away from each other (for active neurons with a firing rate exceeding 3 spikes/ bin).

- (4) *Global alignment across datasets:* To understand how well the populations are aligned in a global sense, we computed the OT and then sorted the rows and columns by their overall firing rate. This reveals already a rough alignment across the datasets with the latent embeddings being roughly organized by firing rates and then with motifs also popping out of the overall OT distance matrix.