# OpenReview forum: "Seeing the forest and the tree: Building representations of both individual and collective dynamics with transformers"
_NeurIPS.cc/2022/Conference — NeurIPS 2022 Accept_

### Official Review · Reviewer_7m6L · 2022-07-11

**Rating:** 8
**Confidence:** 4
**Soundness:** 4 excellent
**Presentation:** 4 excellent
**Contribution:** 4 excellent

**Summary:**

Authors introduce a method to decompose the representations of a population of neurons of the individual dynamics from the population dynamics which allows decoding generalization to unseen populations of neurons.

The results are presented on a behavioral set of benchmark datasets where the author's method outperforms (by a meaningful margin) on 6/10 tasks.

Overall, the work is impactful in that generalization to unseen populations of neurons is a big unblock for the neuroscience community.

**Questions:**

- there wasn't a lot on related work on the comp neuro side. are there related papers that attempt something similar?

**Limitations:**

Have not addressed society impacts directly... however, this is HUGELY (positively) impactful for applications like prosthesis where the prosthesis design needs to generalize to different neural populations for each individual.

**Strengths And Weaknesses:**

Strengths:
- motivation is very strong
- the approach is well explained by breaking up the dynamic into f, g which enable the decomposition desired and a general approach where anything can be used for f or g
- formulation is simple which is a huge plus.

Weaknesses:
- would have liked to see more benchmarks, but understand that there is limited availability of this.

---

> ### Author Response · Authors · 2022-08-02
> **Reply to Reviewer 7m6L**
>
> Thank you for your encouraging words and insightful comments!
>
> **1. “Would have liked to see more benchmarks, but understand that there is limited availability of this.”**
>
> When designing our experiments, we aimed to get a good spread of different tasks and introduce datasets that could challenge different aspects of our model and design goals. We are excited about the promise of the model in other domains and hope that with the rise of more benchmarks and datasets in the community, we can apply our model to novel applications and data in the future.
>
> **2. More “related work on the comp neuro side”**
>
> Combining datasets, and especially learning representations of large neural recordings that can be compared across sessions or even subjects, is an active research area in computational neuroscience. For example, subject-invariant representations can be constructed using adversarial learning to discourage variability across subjects/trials [1, 2] (see Dabagia et al. [3] for a recent review). We will update the related work to situate our paper in this context. Our model is innovative in this respect because it explicitly separates dynamics from channel interactions; thus, the invariance is baked into the architecture itself and does not have to be learned through training over large amounts of data and datasets. This means it can learn transferable representations even when trained on a single dataset.
>
> **3. “Have not addressed society impacts directly... however, this is HUGELY (positively) impactful”**
>
> Thank you for your positive evaluation! Indeed, we think that our method can have positive societal impacts in many ways. For example, it can enable the systematic analysis and tracking of neurodegenerative diseases over time and across individuals, thus helping researchers better understand these diseases. It also provides a more robust framework for brain decoding, which would have great utility for prosthesis interfacing. We will include discussion related to societal impact during revision.
>
> ---
> **References**
>
> [1] Cheng, J. Y., Goh, H., Dogrusoz, K., Tuzel, O., & Azemi, E. (2020). Subject-aware contrastive learning for biosignals. arXiv preprint arXiv:2007.04871.
>
> [2] Gonschorek, Dominic, et al. "Removing inter-experimental variability from functional data in systems neuroscience." Advances in Neural Information Processing Systems 34 (2021): 3706-3719.
>
> [3] Dabagia, Max, et al. "Comparing high-dimensional neural recordings by aligning their low-dimensional latent representations." arXiv preprint arXiv:2205.08413 (2022).

---

### Official Review · Reviewer_Cc4d · 2022-07-11

**Rating:** 8
**Confidence:** 3
**Soundness:** 4 excellent
**Presentation:** 4 excellent
**Contribution:** 4 excellent

**Summary:**

The authors present a novel approach for inference over collective systems, through a separable transformer-based neural network architecture. The novelty lies in learning both the individual components of the system (the trees) as well as the population-level dynamics (the forest). The authors correctly identify that separating individual from system level dynamics has previously been investigated (e.g., in the GroupFormer archetecture [Li, Shuaicheng, et al]) but set themselves apart from previous work by noting that they learn representations at both scales in a quasi-independent manner or, in the authors words, they 'restrain the direct computation' between the two learned maps. To move between the two maps, population and individual, a projection module is learned simultaneously which provides access to both representations.

Crucially, by disentangling learning over the collective into two independent representations, the authors demonstrate that they can use embeddings of individual-scale dynamics for transfer learning on new datasets. This can be applied to new datasets of arbitrary size and/or ordering due to the permutation-invariance property (a consequence of separating individual and population learning tasks).

The authors test their approach on two datasets: a synthetic many-body simulation (of two/three masses in orbit) and neuronal behavioural data. They find strong performance both on the synthetic dataset and on the neuronal data. Importantly, the authors find that they can use a model trained on one dataset and apply the invariant portion of this model to a new unseen dataset. This retrained model outperforms many others which learn from scratch.

Li, Shuaicheng, et al. "Groupformer: Group activity recognition with clustered spatial-temporal transformer." Proceedings of the IEEE/CVF International Conference on Computer Vision. 2021.


**Questions:**

The authors mention in their conclusion that by treating individual systems separately, the individuals have limited dimensionality. I would have also expected that approaches that decouple learned representations of individual from the population have poorer performance on population-scale prediction tasks. My reasoning is that population-based learning can be improved by leveraging/constraining learned representations on accurate relational/individual representations. Do the authors have any thoughts on this?

I believe that decoupling the learning of individual from population allows more efficient scaling to large datasets. Is this correct?

**Limitations:**

The authors highlight two limitations of their work: reliance on deterministic trajectories at the individual-scale and lost information due to treating systems independently. I have asked for clarification on whether population-based predictions also suffer due to decoupling of learning on individuals from population, and whether this is an additional limitation of the approach.




**Strengths And Weaknesses:**

This paper is well structured and written, provides a novel approach to an interesting problem and is considered in its discussion of its own strengths and weaknesses. I think the work will be seen as significant, especially given its framework with respect to challenges in neuroscience.

Specifically, by focussing on learning representations of individual dynamics distinct from population dynamics, the authors rightly point to the added benefit of transfer learning by levering permutation-invariance of the separated system. I see this form of transfer learning as the central result of the approach and the authors did well to emphasis this.

My main concern is that the synthetic dataset is a relatively simple example when compared against the application of neuronal data. The two-body system will not have particularly complex dynamics and I did not find it particularly compelling. Whereas I appreciate the three-body system will have chaotic dynamics under some parameter values, it wasn't clear to me what proportion of the orbits were expected to be chaotic. Additionally, predicting the trajectories and underlying parameters for a three-body system seems an easy problem when compared to the neural dataset. I suggest the authors introduce a third example of intermediate difficulty, either with significantly more individuals/channels, time-varying interactions, or a typical dataset from previous work on neural relational inference (see, for example, [Kipf, Thomas, et al.] and [Graber, Colin, and Alexander Schwing]).

There are some minor languages points. I would consider rewording sentence on lines 86-87. Additionally, the second challenge in section "Learning across recordings or individuals" is not especially clear. I think the authors are referring here to heterogeneity in labels, both within (e.g. a brain), across a single species/system, and across different species/systems. I think the reference to number of neurons (on line 98) is referencing number of channels/individuals.

Kipf, Thomas, et al. "Neural relational inference for interacting systems." International Conference on Machine Learning. PMLR, 2018.

Graber, Colin, and Alexander Schwing. "Dynamic neural relational inference for forecasting trajectories." Proceedings of the IEEE/CVF Conference on Computer Vision and Pattern Recognition Workshops. 2020.

---

> ### Author Response · Authors · 2022-08-02
> **Reply to Reviewer Cc4d**
>
> Thank you for your encouraging evaluation and insightful comments and questions!
>
> **1. “The synthetic dataset is a relatively simple example when compared against the application of neuronal data.”**
>
> Thank you for your insightful suggestion. We agree that the two-body system is an easy one; we believe the three-body system has intermediate difficulty as all of them should be intrinsically chaotic, where the solutions are only obtainable via numerical methods. However, we do select the initial parameters such that they are relatively stable at the beginning.
>
> With this said, we do agree that when comparing with the neural systems, the synthetic experiments are all relatively simple. We would like to thank you for the synthetic datasets recommendations, and we will revise the paper to add them as additional references for future readers. However, we stand by the choice to focus more on extending neural data to different tasks instead of performing more evaluations on complex synthetic datasets.
>
> **2. “Whether population-based predictions suffer due to decoupling of learning on individuals from population, and whether this is an additional limitation of the approach.”**
>
> While the decoupling does not seem to impact the performance of our experiments as indicated by our evaluation results, we do agree that the performance might be affected in certain cases. Actually, we believe the effects of decoupling are two-fold:
> - When the interaction between individuals across different timestamps ($x_{i_1, j_1}$ and $x_{i_2, j_2}$ when $i_1 \neq i_2$ and $j_1 \neq j_2$) are limited or structured, decoupling their representations would reduce unhelpful attention parameters, and thus might help with overfitting. Moreover, modeling individual behaviors before incorporating population-level information might have additional benefits due to the disentanglement.
> - However, we can imagine that decoupling might be harmful in certain cases, e.g. when the individual behavior has limited pattern and is guided by population-level factors in a completely stochastic way. In this case, modeling individuals might give a fragile representation and thus the resulting population-based prediction might suffer.
>
> We think that is an insightful question to ask, and the arguments above are mainly our conjecture without theoretical/empirical justification.
>
> **3. “I believe that decoupling the learning of individual from population allows more efficient scaling to large datasets. Is this correct?”**
>
> Indeed, there are two key computational advantages that come from processing the individual channels independently [1, 2]. 1) We are processing the features across a single dimension (temporal) which means that the first layers of our architecture are computationally more efficient, this is similar to the use of depth-wise convolutions in CNNs for example. 2) We are taking advantage of “weight sharing” because we use the same encoder across channels, this means that when we scale to large populations, we won’t require as much data. While we haven’t explored the potential scalability of our method in this work, we do agree that our framework is better suited for efficient scalability.
>
> **4. Minor language points**
>
> Thank you for the very careful reading and detailed suggestions, we will make sure we edit the language accordingly during revision.
>
> ---
> **References**
>
> [1] Kaplan, J., McCandlish, S., Henighan, T., Brown, T. B., Chess, B., Child, R., ... & Amodei, D. (2020). Scaling laws for neural language models. arXiv preprint arXiv:2001.08361.
>
> [2] Liu, Y., Sangineto, E., Bi, W., Sebe, N., Lepri, B., & Nadai, M. (2021). Efficient training of visual transformers with small datasets. Advances in Neural Information Processing Systems, 34, 23818-23830.

---

### Official Review · Reviewer_6wnv · 2022-07-11

**Rating:** 5
**Confidence:** 4
**Soundness:** 2 fair
**Presentation:** 3 good
**Contribution:** 2 fair

**Summary:**

This paper presents a transformer architecture-based time-varying data learning method. It consists of a temporal-spatial module and a projection module to learn the descriptions of both the individuals and collective population and then leverages the interactions to predict the target behavior. The experimental evaluation is performed on synthetic many-body systems and real neural response datasets.


**Questions:**

Apart from the presentation issues mentioned in the above weak point summary, comparing the Transformer and conventional sequence models on univariate time series in the proposed framework would be useful for assessing the significance of the proposed method.

**Limitations:**

No potential negative societal impact is found.

**Strengths And Weaknesses:**

--- Strong points:

This paper proposes a disentangled representation learning which models individual channel time series and their interactions.


--- Weak points:

The proposed method is built on top of existing ideas and lacks novelty and insights. Meanwhile, some design choices are not well justified theatrically or experimentally.

(a) The problem formulation is confusing. As is mentioned in Sec. 3.1, the input observation and target sequence are both in the time range from 1 to T. Depending on the conditioned observations on which the prediction of the target at a certain timestamp is, it can lead to different types of problem setups, e.g., forecasting, filtering or smoothing. These problems have different directions of solutions and the author is expected to explicitly specify the problem type.

(b) The optimization problem (3) is ill-defined in that the input into g() is of the shape N*T, while the function g() is defined as a mapping from R^N to R^N. Meanwhile, this definition is different from the loss function (4), thereby giving the impression that the optimization problem seems not clearly defined in the first place and the subsequently proposed method can hardly be solid enough.

(c) The temporal-spatial attention and then slicing along different dimensions of the output representations have been extensively studied in the architectures like RNN and CNN [1, 3]. It is also not novel to explore disentangled individual representations of multi-channel time series [2, 3]. The Transformer is applied in this paper as an off-the-shelf encoder. Overall, the paper seems to bear marginal technical contributions regarding model architectures.

[1] Qin, Yao, et al. "A dual-stage attention-based recurrent neural network for time series prediction." Proceedings of the 26th International Joint Conference on Artificial Intelligence. 2017.

[2] Guo, Tian, Tao Lin, and Nino Antulov-Fantulin. "Exploring interpretable LSTM neural networks over multi-variable data." International conference on machine learning. PMLR, 2019.

[3] Assaf, Roy, et al. "Mtex-CNN: Multivariate time series explanations for predictions with convolutional neural networks." 2019 IEEE International Conference on Data Mining (ICDM). IEEE, 2019.

(d) The motivation for using the Transformer on individual time series is weak. The Transformer is intended to capture pair-wise relations between tokens/steps in a multi-dimensional sequence. As far as I know, recent works mostly applied the Transformer or variants to multi-dimension time series [4, 5]. However, applying Transformer on univariate time series riks overcomplicating the model and overfitting as well. It would be more convincing to see the comparison between conventional sequence models and Transformer in the proposed framework.

[4] Tang, Binh, and David S. Matteson. "Probabilistic transformer for time series analysis." Advances in Neural Information Processing Systems 34 (2021): 23592-23608.

[5] Zhou, Tian, et al. "FEDformer: Frequency enhanced decomposed transformer for long-term series forecasting." arXiv preprint arXiv:2201.12740 (2022).

(d) A projection module and two different project operations are described in Sec. 3.2. It is necessary to clarify their difference and connection. Meanwhile, the projection module consists of some plain operations, e.g., dense transformation and concatenation. Technically, this module bears marginal novelty or new insights.

(e) In the synthetic data experiment, there is no reference for the baseline method and thus it is difficult to evaluate the significance of the proposed method.

(f) The hyper-parameter search process is missing for baselines. It is important to describe the process, since it significantly affects the performance and the validity of the comparison.

---

> ### Author Response · Authors · 2022-08-02
> **Reply to Reviewer 6wnv**
>
> Thank you for your insightful feedback. We hope both the formulation and the significance of the work will be clarified through our response below.
>
> ### **Clarification of the problem formulation**
> **1. “The “problem setups” is unclear, “e.g., forecasting, filtering or smoothing.” … “The author is expected to explicitly specify the problem type.”**
>
> In addition to providing formal definitions of the data and labels (see lines 147-148) in the methods and in Eq (3) and (4), in lines 188-189, we state that “$l$ can be set to either classification loss (CrossEntropy) or regression loss (MSE), depending on the type of prediction target we consider”. In all of our experiments, we specify whether a time-varying classification (Table 2) or regression task (Table 1, 2, 3) is considered.
>
>
> **2. “The optimization problem (3) is ill-defined in that the input into g() is of the shape N*T, while the function g() is defined as a mapping from R^N to R^N.”**
>
> We would like to point out that the input into $g()$ is not of shape N*T for equation 3. The input to $g()$ is $(f(X^{(1)})_j, …, f(X^{(N)})_j)$, where the subscript notation $j$ is used to denote the jth element of $f(X^{(i)}) \in \mathbb{R}^T$, making the input to $g$ of size $N$. We will make this more clear in the revised paper.
>
> **3. Clarification regarding the differences between Eq (3) and (4)**
>
> Equation 3 is a generalized/simplified form of the framework that we proposed, while Equation 4 is a special case of Equation 3 that we get when we select specific modules, and incorporate a multi-stage loss which provides further training flexibility. Specifically:
>
> - The LHS of equation 4 is equation 3 with customized modules: In $\sum_{j=1}^{T} \ell (\operatorname{Proj-Int}([v_{1 j}, \ldots, v_{N j}]), y_j)$, the $v_{i j}$ ($\widehat{v}_{i j}$ in original text) is the product produced by the individual transformer $f()$ as well as the interaction transformer $g()$. This is stated in line 165 and line 174.
> - In line 184 we begin to introduce our multi-stage loss, where the loss is a summation of the LHS of equation 4 and an additional part (the RHS of equation 4), where we explained our motivation in line 184-186 and line 190-195.
>
> We will add a sentence in the remark to explicitly state their connections to clarify.
>
> ### **Novelty of our work**
>
> **4. “The temporal-spatial attention and then slicing along different dimensions … have been extensively studied … It is also not novel to explore disentangled individual representations of multi-channel time series.”**
>
> While it is true that there are numerous works in video and multivariate time-series forecasting where temporal-spatial attention is used [1, 2] and the effects of individual channels are interpreted [3, 4], most of these models are still trained on all of the data jointly with all of the channels being fed in collectively from the beginning. This means that any attention they learn over the data and the inputs cannot be easily transferred in a new condition with a different size population and/or a different permutation of the input.
>
> Different from them, by building our model in a separable way, we can apply it to new populations of arbitrary size and ordering, and this is what enables transfer in our model. We will also revise our related work section to further clarify this point.
>
> **5. “The motivation for using the Transformer on individual time series is weak… Applying Transformer on univariate time series risks overcomplicating the model and overfitting as well. It would be more convincing to see the comparison between conventional sequence models and Transformer in the proposed framework.”**
>
> We applied transformers as our building blocks because of their versatility, high expressivity, and lack of strong biases when compared with RNN/LSTM. This is also true for the interaction module where we have no prior knowledge about the interactions between channels, so the transformer’s attention mechanism is advantageous.
>
> We agree that it would be useful to evaluate how these different models would perform with the proposed framework. In the context of this work, however, we find that the selected model yields compelling results. As for the potential overfitting issue for applying transformers, in the datasets that we tested (Table 2 and Table 3 in the paper), we show that our model’s performance doesn’t suffer from overfitting issues and instead learns a rich representation of each neuron’s activity that allows for linear readouts and decoding with high accuracy. Our individual representation is robust even when doing generalization/transfer experiments when the model is applied to a new population.

---

> ### Author Response · Authors · 2022-08-02
> **Reply to Reviewer 6wnv -- part 2**
>
> ### **Other questions**
> **6. “A projection module and two different project operations are described in Sec. 3.2. It is necessary to clarify their difference and connection.”**
>
> We described two transformer modules, and one projection module in Sec. 3.2. The transformer module is used to learn either a temporally-informed representation (see line 160-162) or a population-informed representation (see line 167-169), while the projection module is used to form a population representation from individual representations (see line 175-176).
>
> **7. “In the synthetic data experiment, there is no reference for the baseline method and thus it is difficult to evaluate the significance of the proposed method.”**
>
> We used the NDT architecture in [7], where the encoder architecture and model complexity remain the same when comparing to our model, but the individual information and population information are not separated. We will make sure we cite it properly during revision.
>
> **8. “The hyper-parameter search process is missing for baselines.”**
>
> Thank you for pointing this out! We will include the hyper-parameter searching procedures during revision in the Appendix. For the mihi-chewie experiments, we did a grid-search of hyper-parameters on the mihi-1 dataset for the decoding task for each model, and used the same parameters throughout. Similar procedures are conducted for synthetic experiments and Maze experiments, but the grid-search is done for each individual task.
>
> ---
> **Reference**
>
> [1] Li, S., Cao, Q., Liu, L., Yang, K., Liu, S., Hou, J., & Yi, S. (2021). Groupformer: Group activity recognition with clustered spatial-temporal transformer. In Proceedings of the IEEE/CVF International Conference on Computer Vision (pp. 13668-13677).
>
> [2] Hsieh, J. T., Liu, B., Huang, D. A., Fei-Fei, L. F., & Niebles, J. C. (2018). Learning to decompose and disentangle representations for video prediction. Advances in neural information processing systems, 31.
>
> [3] Guo, T., Lin, T., & Antulov-Fantulin, N. (2019, May). Exploring interpretable lstm neural networks over multi-variable data. In International conference on machine learning (pp. 2494-2504). PMLR.
>
> [4] Assaf, R., Giurgiu, I., Bagehorn, F., & Schumann, A. (2019, November). Mtex-cnn: Multivariate time series explanations for predictions with convolutional neural networks. In 2019 IEEE International Conference on Data Mining (ICDM) (pp. 952-957). IEEE.
>
> [5] Dosovitskiy, A., Beyer, L., Kolesnikov, A., Weissenborn, D., Zhai, X., Unterthiner, T., ... & Houlsby, N. (2020). An image is worth 16x16 words: Transformers for image recognition at scale. arXiv preprint arXiv:2010.11929.
>
> [6] Arnab, A., Dehghani, M., Heigold, G., Sun, C., Lučić, M., & Schmid, C. (2021). Vivit: A video vision transformer. In Proceedings of the IEEE/CVF International Conference on Computer Vision (pp. 6836-6846).
>
> [7] Ye, J., & Pandarinath, C. (2021). Representation learning for neural population activity with Neural Data Transformers. arXiv preprint arXiv:2108.01210.

---

> > ### Comment · Reviewer_6wnv · 2022-08-09
> > **Reply to authors' answers**
> >
> > Thanks for the detailed answers!
> >
> > Regarding the problem formulation and some presentation issues, the authors mostly resolve the concern. The novelty of the work is also clarified through the answer and the general response.
> >
> > Although the author claims the novelty in terms of the general framework, especially for neuroscience, the implementation of each framework component is based on standard model architectures, loss functions, and optimization processes. This implementation renders the framework still lacking some new insights, in my opinion.
> >
> > Alternatively, if the novelty of the framework is spefiic for certain domains, e.g., neuroscience, it would be better to differentiate the work from the typical setting of multivariate time series in the problem formulation. In the current version of the paper, the problem was defined in a relatively ordinary way in multivariate time series analysis and insufficiently stressed the specificity of the problem in the context of neuroscience or others.
> >
> > Overall, given the author's response, I would like to increase the rating to "Borderline accept".

---

### Official Review · Reviewer_xZtG · 2022-07-11

**Rating:** 5
**Confidence:** 4
**Soundness:** 4 excellent
**Presentation:** 3 good
**Contribution:** 2 fair

**Summary:**

The paper presents a neural architecture that models a dynamical system by building representations for both individual and collective population dynamics separately. The first transformer operates only on individual time series and the second transformer acts as an interaction network to come up with the collective representation of the models. The models are trained to classify or regress the target values. Authors then test this model on a toy datasets and neural activity modelling.

**Questions:**

- The method doesn't seem to infer or decompose entities from raw observational data. It requires the knowledge of separate channels and feed them separately to individual transformer which a simpler problem than a cocktail party problem. Doesn't it restrict its applicability for e.g. in many cases we don't know the number of interacting agents.
- How do you infer the components/objects/neurons whose dynamics needs to be modelled individually?
- In the works of [1], [2], [3], [4] the models first try to infer individual entities from the observed frames in videos and then try to model dynamics of each entity individually. Moreover, an interaction module is used to explicitly model objects interaction and combined influence on their dynamics. [3] even uses transformer for modelling dynamics. It would be interesting to position this paper w.r.t them.
- Can interaction transformer introduce some kind of bias towards the order of channels?
- Since it’s not a generative model, it’s hard to quantitatively see how individual channels are independent. Could there be a test to measure how independent individual dynamics are? Can authors try permuting the channels and see if the results on the test set remain the same?
- [Related to above question]: Did you try dropping one or a few channel to see how models behave?
- How does interaction network $g$ capture complex interaction without any specific inductive bias for capturing such interactions for e.g. as shown in [5]. - Could there be any specific interaction module to account for the interactions between individual agents? Would it work for physics based interactions between agents e.g. collision between objects or if agents exhibits some magnetic push/pull?

[1] https://arxiv.org/abs/1806.04166
[2] https://arxiv.org/abs/1806.01794
[3] https://arxiv.org/abs/2107.09240
[4] https://arxiv.org/abs/2205.14065
[5] https://arxiv.org/abs/1612.00222


**Limitations:**

The model applicability seems to be limited to the problems where the exact number of agents contributing to overall population dynamics need to be known beforehand.

**Strengths And Weaknesses:**

**Strengths**
- The paper is well written and easy to follow.
- The paper demonstrates the clear benefits of decomposing dynamics modelling into  individual dynamics and collective dynamic for better transfer learning.
- The application to neural activity datasets is interesting.

**Weaknesses:**
- The method is limited in terms of novelty as the individual modules for capturing individual dynamics and collective dynamics are commonly used transformer architectures.
- The method is only applicable to the data where the existence and number of separate channels in the input data is known. The limits it applicability to models where the number of interacting agents in the systems are not known.

---

> ### Author Response · Authors · 2022-08-02
> **Reply to Reviewer xZtG**
>
> We would like to thank you for your constructive feedback. Aside from emphasizing the novelty of our work in the general response, we would like to answer your questions as below:
>
> ### **Novelty of our work**
> **1. “The method is limited in terms of novelty as the individual modules … are commonly used transformer architectures.”**
>
> As we note in the general response, a core innovation of this work is learning individual-level representations and decomposing this computation from the population-level transformations. We agree that some of the components we selected are somewhat standard; however, we believe the novelty of the work is still significant because: 1) the goal of our work is not in developing new custom components, but in designing a simple and generalizable framework that addresses inherent challenges in many fields (e.g. neuroscience); 2) Our specific design of the implementation, training objective (a flexible multi-stage loss), and the functional alignment method we proposed still has novelty over previous works. Moreover, our architecture and the corresponding evaluation methods are particularly of great utility in neuroscience, as it enables the systematic comparison of neural states in longitudinal or cross-sectional studies. This helps us to move towards foundational models where we can train over large corpora of data from different individuals.
>
> ### **Applying the method to observations from mixtures of sources**
> **2. “The method is only applicable to the data where the existence and number of separate channels in the input data is known. This limits its applicability …” “The method doesn't seem to infer or decompose entities from raw observational data.”**
>
> While it is true that our method is not designed to decompose different sources from raw observational data directly, we don’t see this as a drawback and still believe that our approach has a lot of utility in a wide range of domains.  The  focus of this work is on learning permutation-invariant representations that can be transferred and are generalizable across different populations of different size and ordering and to do this, we need to make certain assumptions on the potential identifiability of transferrable dynamics. Without first separating sources in raw observational data, our problem of transfer would be ill-defined.
>
> At the same time, we see the general mechanism and model that we have developed as powerful enough to deal with poorly decomposed channels that may have noise or some corruptions within. This would be an interesting avenue for future research.   We plan to include discussion about dealing with data from a mixture of sources as a potential limitation and direction for future work.
>
> **3. “How do you infer the components/objects/neurons whose dynamics need to be modeled individually?”**
>
> For neural activities, we obtain the data after spike sorting (mentioned in line 289-291 and details in Supp Materials Section 3.1.1). In this case, a clustering problem is solved to assign spikes to specific neurons and then from those spike times, we then compute a firing rate for each neuron at each point in time (100 ms time bins).  Spike sorting is a standard method applied to pre-process electrophysiological recordings from populations of neurons [1].
> We will include a further detailed explanation of the pre-processing of neural activity data used in our experiments inside the Appendix.
>
> ### **Relationship with related works**
> **4. “Position this paper with respect to” related works**
>
> Thank you for pointing us to related papers in object-centric representation learning methods in vision [2, 3, 4, 5] and complex systems [6]. As mentioned in the general response, we definitely will include the suggested works in the revised related work section. However, we would like to point out that our method is different from these approaches in how it decouples temporal and spatial information: While there are numerous works in video or multivariate time-series where temporal-spatial attention is applied to separate or interpret the objects within, most of these models are still trained on data where all objects/individuals exist are passed in jointly at the input. This means that the attention they learn over the data and the inputs cannot be easily transferred in a new condition where the input is passed in with a different ordering (as in neuroscience). Learning transferable representation of individual objects that is independent of modality and prior knowledge of interactions is one of our key contributions.

---

> ### Author Response · Authors · 2022-08-02
> **Reply to Reviewer xZtG -- part 2**
>
> ### **Other questions**
> **5. “Can an interaction transformer introduce some kind of bias towards the order of channels?” “Can authors try permuting the channels and see if the results on the test set remain the same?”**
>
> Our results in transfer across days and different animals tests an extreme case where the model operates on different  sets of neurons of different orders and sizes; In these experiments, we provide compelling evidence that our model can learn rich features from data independent of the  ordering of the channels. Moreover, we find that transfer is possible across populations of different sizes; note that we use 163, 148, 163, 152 neurons for C-1, C-2, M-1, M-2, respectively, where each set has uncorrelated ordering and non-overlapping sets.
>
> To better understand the sensitivity of the method to channel ordering, we repeated our experiments with the whole model re-trained based on a permuted ordering of neurons for five different random seeds. The resulting mean accuracy and standard deviation are computed as below, and is benchmarked with the numbers of the same random seed but without the permuted ordering:
>
> |      |   C1   |   C2   |   M1   |   M2   |
> |---------|-------|-------| -------| -------|
> |Without perm| 79.86$\pm$1.23 |  82.13$\pm$0.52  |  86.83$\pm$0.76   |   80.41$\pm$0.92   |
> |With perm|  80.22$\pm$1.79   |   81.84$\pm$1.88   |   85.78$\pm$0.98   |   80.05$\pm$1.70   |
>
> As we can see, the permutation of the input channels does not have a huge impact on the performance overall, but did increase the standard deviation of the performance by a small margin. We will include the results as well as the additional discussion in the Appendix. Thank you for your suggestion!
>
> **6. Did you try dropping one or a few channels to see how models behave?**
>
> Thank you for raising this question. During training we use channel dropout to build robustness into the model, however, we haven’t yet investigated the performance of the model when tested on data with dropped channels. This would be an interesting followup study to investigate how the information is distributed among the neuron population.
>
> **7. “Could there be any specific interaction module to account for the interactions between individual agents? Would it work for physics based interactions between agents?”**
>
> We thank you for this insightful question. We do think that replacing the transformer encoders to more specific architecture that is accountable for specific types of interactions is a valuable line of future work. In this work, we just applied a versatile architecture, but certainly there exists many different models (e.g. Graph-based encoders [7, 8], RNN/LSTM encoders [9]) that might perform better in different cases. We will include a paragraph of limitation/future work during revision to make sure we discuss this.
>
> ---
> **References**
>
> [1] Quian Quiroga R, Nadasdy Z, Ben-Shaul Y (2004) Unsupervised Spike Detection and Sorting with Wavelets and Superparamagnetic Clustering. Neural Comp 16:1661-1687.
>
> [2] Hsieh, J. T., Liu, B., Huang, D. A., Fei-Fei, L. F., & Niebles, J. C. (2018). Learning to decompose and disentangle representations for video prediction. Advances in neural information processing systems, 31.
>
> [3] Kosiorek, A., Kim, H., Teh, Y. W., & Posner, I. (2018). Sequential attend, infer, repeat: Generative modelling of moving objects. Advances in Neural Information Processing Systems, 31.
>
> [4] Wu, Y. F., Yoon, J., & Ahn, S. (2021, July). Generative Video Transformer: Can Objects be the Words?. In International Conference on Machine Learning (pp. 11307-11318). PMLR.
>
> [5] Singh, G., Wu, Y. F., & Ahn, S. (2022). Simple Unsupervised Object-Centric Learning for Complex and Naturalistic Videos. arXiv preprint arXiv:2205.14065.
>
> [6] Battaglia, P., Pascanu, R., Lai, M., & Jimenez Rezende, D. (2016). Interaction networks for learning about objects, relations and physics. Advances in neural information processing systems, 29.
>
> [7] Guo, S., Lin, Y., Feng, N., Song, C., & Wan, H. (2019, July). Attention based spatial-temporal graph convolutional networks for traffic flow forecasting. In Proceedings of the AAAI conference on artificial intelligence (Vol. 33, No. 01, pp. 922-929).
>
> [8] Xu, M., Dai, W., Liu, C., Gao, X., Lin, W., Qi, G. J., & Xiong, H. (2020). Spatial-temporal transformer networks for traffic flow forecasting. arXiv preprint arXiv:2001.02908.
>
> [9] Guo, T., Lin, T., & Antulov-Fantulin, N. (2019, May). Exploring interpretable lstm neural networks over multi-variable data. In International conference on machine learning (pp. 2494-2504). PMLR.

---

### Author Response · Authors · 2022-08-02
**General Response**

We would like to thank all reviewers for their insightful feedback and questions. We greatly appreciate the overall enthusiasm for the approach and the reviewer’s assessment of the significance of our work as *“a big unblock for the neuroscience community”* and *“HUGELY (positively) impactful”* (Reviewer 7m6L); the assessment of the methodology as *“novel approach to an interesting problem”* (Reviewer Cc4d) with *“clear added benefits … for better transfer learning”* (Reviewer xZtG); and the presentation of the work as *“well-written”* (Reviewer xZtG), *“well structured”* (Reviewer Cc4d), and *“well explained”* with a *“simple formulation”* (Reviewer 7m6L).

In the rest of this general response, we would like to address some of the reviewer comments, discuss the novelty of the work, and clarify some of our design decisions.

### **Core innovation**
While some reviewer comments praised our framework and highlighted our “quasi-independent” (Reviewer Cc4d) learning framework, other reviewer comments that focused purely on the architecture skipped over what we believe to be a critical contribution of the work.

In particular, we see a core innovation of our method is in its ability to form individual-level representations that are 1) invariant to the ordering and size of the population and 2) can be generalized/transferred to unseen populations of a new set of agents/sources, or of new conditions. This is a direct result of the design of our architecture which restricts the direct computation between representations formed at the level of single channels and those formed at the population level (line 101-104). We believe that our proposed framework provides a simple yet elegant solution to problems where a representation that generalizes to unseen channels is required, which appears in many fields (e.g. neuroscience).

### **How it differs from related works**
While there are many works in video and multivariate time-series where temporal-spatial attention is applied to separate or interpret the objects within, most of these models are still *trained on all of the data jointly, with all of the channels being fed in collectively at the beginning*. This means that any attention they learn over the data and the inputs cannot be easily transferred when the channels are shuffled or in a new condition with a different population of arbitrary size. This is true for the related works pointed out by the reviewers, both in terms of work that (a) learns from multivariate time series data in an interpretable way [1, 2, 7]; and (b) unsupervised object-centric representation learning approaches used in vision [3, 4, 5, 6].

### **Novelty of architecture**
Reviewer xZtG and 6wnv raised some concerns about the novelty of our architecture because we used relatively standard transformer encoders as building blocks. While it is true that we do not apply more advanced extensions to our transformer backbones, the goal of our work is not in developing new custom components, but in designing a simple and generalizable framework that addresses inherent challenges in many fields (e.g. neuroscience). We use transformers as our building block because of their versatility, high capacity for learning, and relatively low inductive bias.

Moreover, our implementation, training objective (a flexible multi-stage loss), and the functional alignment method that we proposed have novelty over previous works. Our post hoc evaluation methods are particularly of great utility in neuroscience, as it enables the systematic comparison of neural states in longitudinal or cross-sectional studies, and helps us to move towards foundational models where we can train over large corpora of data from different individuals.

### **Planned revisions: additional related works and discussions of limitations and impact**

Again, we would like to thank all reviewers for their valuable feedback. Based upon this feedback, we identified the following high-level areas for improvement:
1) Introducing additional related works and discussing their difference from our work;
2) Stating additional directions for future works, especially regarding replacing the encoder with different architectures for different applications;
3) Discussing additional limitations and the social impact of our work.

Thus, we plan to revise the manuscript by broadening the related work section as discussed above, and adding a section under the conclusion section to address limitations/future works from reviewers’ feedback. We would like to thank the reviewers for their constructive feedback that helped us improve the manuscript.

---

> ### Author Response · Authors · 2022-08-02
> **General Response References**
>
> **References**
>
> [1] Guo, T., Lin, T., & Antulov-Fantulin, N. (2019, May). Exploring interpretable lstm neural networks over multi-variable data. In International conference on machine learning (pp. 2494-2504). PMLR.
>
> [2] Assaf, R., Giurgiu, I., Bagehorn, F., & Schumann, A. (2019, November). Mtex-cnn: Multivariate time series explanations for predictions with convolutional neural networks. In 2019 IEEE International Conference on Data Mining (ICDM) (pp. 952-957). IEEE.
>
> [3] Hsieh, J. T., Liu, B., Huang, D. A., Fei-Fei, L. F., & Niebles, J. C. (2018). Learning to decompose and disentangle representations for video prediction. Advances in neural information processing systems, 31.
>
> [4] Kosiorek, A., Kim, H., Teh, Y. W., & Posner, I. (2018). Sequential attend, infer, repeat: Generative modelling of moving objects. Advances in Neural Information Processing Systems, 31.
>
> [5] Wu, Y. F., Yoon, J., & Ahn, S. (2021, July). Generative Video Transformer: Can Objects be the Words?. In International Conference on Machine Learning (pp. 11307-11318). PMLR.
>
> [6] Singh, G., Wu, Y. F., & Ahn, S. (2022). Simple Unsupervised Object-Centric Learning for Complex and Naturalistic Videos. arXiv preprint arXiv:2205.14065.
>
> [7] Battaglia, P., Pascanu, R., Lai, M., & Jimenez Rezende, D. (2016). Interaction networks for learning about objects, relations and physics. Advances in neural information processing systems, 29.

---

### Meta-Review · Area_Chair_AgVb · 2022-08-23

**Recommendation:** Accept
**Confidence:** Less certain

**Metareview:**

This paper introduces the multi-stage Embedded Interaction Transformer which models individual channels of systems with multiple interacting elements and then models their interactions.  The approach is applied to simulated systems to recover known interaction-dynamics as well as neural datasets, revealing transferability of some model parameters across animals.  Essentially the model is structured to solve a supervised regression/classification problem where the input corresponds to many (possibly a variable number of) timeseries and the timeseries channels reflect observations of interacting elements.

Reviewers generally found the paper clear, proposing a simple innovation, and applied to an interesting problem class involving neural data.  Two of the reviewers expressed some concerns about the degree of novelty in the proposed approach.  The authors responded to this point directly and while they didn't totally satisfy reviewer 6wnv, this reviewer still updated their rating from a 4 to a 5.

In my own assessment, aligned with the less enthusiastic reviewers, I found the technical contribution somewhat incremental, but clearly enough described.  And I found the evaluation somewhat limited (consistent with reviewer 7m6L), given the moderate magnitudes of improvement, but adequate.  That the EIT generalizes better than other approaches does increase its potential impact and may inspire future follow-up work.  Given the distribution of reviewer ratings, I'm willing to endorse the paper.

**Award:**

No

---

### Decision · Program_Chairs · 2022-09-14

Accept